# 3D Computational Fluid Dynamics Analysis of a Convective Drying Chamber

Miguel Andrés Mauricio Daza-Gómez [1,*], Carlos Andrés Gómez Velasco [2], Juan Carlos Gómez Daza [2] and Nicolás Ratkovich [1,*]

1 Department of Chemical and Food Engineering, Universidad de los Andes, Bogotá 111711, Colombia
2 Food Engineering School, BioNovo, Universidad del Valle, Santiago de Cali 760042, Colombia
* Correspondence: ma.daza299@uniandes.edu.co (M.A.M.D.-G.); n.rios262@uniandes.edu.co (N.R.)

**Abstract:** The drying industry has grown considerably due to the tremendous demand for non-perishable food. Convective drying is one of the most popular equipment in the drying industry (food, chemical, pharmaceutical, etc.). One of the drawbacks of this equipment, when used for convective drying, is the non-uniformity in the final product quality. This study presents the development of a numerical model through Computational Fluid Dynamics (CFD). The drying chamber of a heat pump dryer is assessed from the perspective of drying air velocity and temperature profiles. The model was developed by solving different transport phenomena-related equations. The established methodology was set up to evaluate how the drying air velocity and temperature distribution affect the drying chamber. These results will define if there is a need to redesign it. The air velocity and temperature profile results show a need to redesign the chamber. Only trays 2, 3, and 4 are the ones that would achieve the drying of the products. The proposed solution is to implement air distributors or modify the tray positioning to make the drying air and temperature distribution homogeneous.

**Keywords:** drying industry; convective drying; CFD; heat pump dryer





## 1. Introduction

The significant demands of the drying industry constantly push the development of new technologies and equipment. In the last decades, considerable efforts have been devoted to understanding the changes in drying operations, aiming to develop different ways to prevent undesirable quality losses [1–10]. Drying not only has applications in the food area, but it also expands to industries such as (bio)chemical, pharmaceutical, and agricultural sectors [11]. Among the equipment available in the drying industry, the heat pump dryer is the most used, which has a potential for heat recovery and relatively high energy efficiency [12]. In heat pump dryers, sensible and latent heat of evaporated water is recovered and recycled by reheating the dehumidified air [13]. Drying processes have facilitated the use of new products due to the easy incorporation of dry products into prepared dishes, snacks, bakery, and pastry products, among others [14]. Typical drying parameters of different fruits and vegetables that can be dried in the observed chamber is presented in Table 1.

**Table 1.** Typical drying parameters of different fruits and vegetables that can be dried in the observed chamber [15].

| Researchers | Material | Year | Ta (°C) | HR (%) | Vel (m/s) | Other variables | Dependent variables | Refrigerant | |
|---|---|---|---|---|---|---|---|---|---|
| Vasquez et al. | Grape | 1997 | 50 | | 3 | Pretto | Kinetics | 134a | Cinética |
| Prasertsan et al. | Banana ABB | 1998 | 50–60 | | | | MER, SMER, COP | R-22 | MER, SMER, COP |
| Rahman et al | Green peas | 1998 | 25–60 | 20–60 | 1.5 | | Kinetics, isotherms | | Cinética, isotermas |
| Chua et al. | Guava | 2000 | var. Cyclic | 20–65 | 2.5 | | Moisture content, ascorbic acid | | Contenido humedad, ácido ascórbico |
| Chua et al. | Banana, guava, potato | 2000a | 20–40 | 20–65 | 2.5 | | Kinetics, color variation | | Cinética, variacion color |
| Chua et al. | Banana, guava | 2002b | 25, 30, 40 | | 2.5 | t-cycle | Drying time, color | | Tiempo secado, color |
| Tia et al. | Pineapple, banana, bean, cabbage | 2000 | 45–55 | | | t, flow | DR, MER, SMER, SEC, COP | R-22 | DR, MER, SMER, SEC, COP |
| Achariyaviriya et al. | Papaya | 2000 | | | | fracc-rec | DR, SMER | | DR, SMER |
| Alves-Filho | Fruits and vegetables | 2002 | 20–30 | | 0.5–1.5 | | Kinetics, color, law, size, density | $CO_2$ | Cinética, color, law, tamano, densidad |
| Cardona et al. | Lactic acid bacteria | 2002 | 10, 15, 20, 25 | | 1.71 | medium | Viability, activity | | Viabilidad, actividad |
| Teeboonma et al. | Papaya, mango glaze | 2003 | 45–55 | | | kg/h air | Optimization: Minimum annual cost | R-22 | Optimización: Minimo costo anual |
| Hawlader et al. | Food grains | 2003 | | | | | COP | R11-R12 | COP |
| Hawlader et al. | Ginger | 2006a | 45 | 10 | 0.7 | atm-mod | Gingerol loss | | Pérdida gingerol |
| Hawlader et al. | Apple, guava, potato | 2006b | 45 | 10 | 0.7 | atm-mod | Color, porosity, rehydration, texture | | Color, porosidad, rehidratación, textura |
| Hawlader et al. | Guava, papaya | 2006c | 45 | 10 | 0.7 | atm-mod | Color, porosity, rehydration, text, Vit C | | Color, porosidad, rehidrat, text, Vit C |
| Ortiz | Banana | 2003 | 10, 60 | | 2 | | Color, aw, drying time, moisture | R-22 | Color, aw, tiempo secado, humedad |
| Sosle et al. | Apple | 2003 | 45–65 | 30–50 | | | Rehydration time, SEC, SMER | R-22 | Tiempo rehidratación, SEC, SMER |
| Kohayakawa et al. | Sliced mango | 2004 | 40, 46, 56 | | 1.6–4.4 | thickness | Effective diffusivity, COP | R-22 | Difusividad efectiva, COP |
| Queiroz et al. | Tomatoes | 2004 | 40, 45, 50 | | 1.5–2 | type | Kinetics | R-22 | Cinética |
| Moreira et al. | Chestnut | 2005 | 45, 55, 65 | 20–40 | 1.8–2.7 | | Kinetic Modeling | | Modelamiento Cinética |
| Sun et al. | Potato | 2005 | 45 | 20 | 1.7 | type of heat | Drying speed, temperature profile | | Velocidad secado, perfil temperatura |
| Fatouh et al | Corcholo, herb, parsley | 2006 | 45, 50, 55 | | 1.2, 1.9, 2.7 | size | SEC, drying characteristics | 134a | SEC, caracteristicas secado |
| Ceylan et al. | Kiwi, avocado, banana | 2007 | 40 | | 0.03–0.39 | | MR, DR | | MR, DR |
| Sunthonvit et al. | Nectarines | 2007 | 25 | 10 | 1.6 | | Volatile compound composition | | Composición compuestos volátiles |
| Xanthopoulos et al. | Fig | 2007 | 46–60 | | 1, 5 | | Single layer drying models | R-22 | Modelos secado capa única |
| Ceylan y Aktas | Hazelnut | 2008 | 40, 45, 50 | | | | Time, air speed | | Tiempo, velocidad aire |
| Shi ét al. | Tuna | 2008 | 10, 40 | | 1, 4 | load | Colour, SMER | 134a | Color, SMER |
| Shi et al. | Tuna | 2008a | 20–30 | | 1.5–2.5 | %NaC1 | SMER, DR, color, TVBN | 134a | SMER, DR, color, TVBN |
| Alves-Filho et al. | Isolated protein | 2008 | −5, 25 | | 1, 2.3 | time | MRR, color, density, shrinkage | | MRR, color, densidad, encogimiento |
| Aktas et al. | Apple | 2009 | | | | t, load | Effective diffusivity, COP, DR | | Difusividad efectiva, COP, DR |
| Erbay YIcier | Olive leaves | 2009 | 45–55 | | 0.5–1.5 | time | Phenolic content, antioxidant act., hum | R407C | Contenido fenólico, act. antioxid, hum |
| Lee and Kim | Radish | 2009 | 40 | | | flow | Time, MER, SMER, energy saving | 134a | Tiempo, MER, SMER, ahorro energia |

A good prediction of convective drying processes can be essential for improving processes and minimising problems. Some examples of these problems are high energy consumption, excessive load and wear on equipment, and low yields (number of products with correct characteristics) [14]. Predicting these processes is challenging because it depends on many factors, such as the air's speed, temperature, humidity, level of turbulence, uniform airflow, etc. Keeping these factors under the right conditions is the most challenging task at a pilot and industrial scale [16]. Temperature and air velocity are the two factors that must be considered since they are difficult to measure due to the large number of sensors placed in the chamber. The drying process result depends on the material's location in the dryer since the drying rate depends on the airflow in the drying chamber [17]. Using a computer simulation of the mathematical model of the operation will allow controlling the dynamics of the drying process. This will optimise the dryer's performance in terms of energy consumption, efficiency, and product quality. The numerical methods are also beneficial in saving both time and money [18–20].

The mathematical models in the drying process have been studied under different conditions and configurations (i.e., porous media and shrinkage) in diverse investigations, most of which are approximated in two dimensions. This has provided a significant compression of the drying phenomenon, but the airflow in a convective dryer is usually turbulent and in three dimensions. Therefore, it is of great interest to perform an analysis of the process of turbulent flow and air distribution in a fully three-dimensional way, which implies a non-negligible load on both computational and in the features and scope of the model [14]. A complete model of the drying process must consider the interaction

between heat and mass transfer within the material to be dried and the transfer to the drying airflow [21]. Several studies have been carried out both by simulations and experimental facilities. These studies include the ones performed by Cârlescu et al. [22], Villegas et al. [23], Lemus-Mondaca et al. [14,24], Han et al. [25], Gómez and Ochoa [26], Ozgen [27], Mohan [16], Lamnatou [21], among many others.

Recent studies have assessed the heat pump dryer with the compressor outside the air circuit and have also investigated powering it through photovoltaic means [28–36]. Other studies have been done with a hybrid approach [37–39]. The experimental facility assessed in this paper has the compressor of the heat pump dryer inside the air circuit. This study aimed to use a three-dimensional Computational Fluid Dynamics (CFD) model of a convective drying chamber to analyse the airflow behaviour in the chamber geometry and its effect on the temperature distribution during the convective drying process. This will provide the basis to redesign the chamber and increase the efficiency of the modified heat pump dryer at Universidad del Valle, Cali, Colombia.

## 2. Materials and Methods

### 2.1. Mathematical Model

The mathematical models used to investigate the flow and heat transfer are momentum and energy equations, turbulence models, and the appropriate boundary conditions. The physical properties of air are not considered constant. The governing equations are presented below in Equations (1)–(3).

Continuity

$$\frac{\partial \rho}{\partial t} + \nabla \cdot (\rho v) = 0 \tag{1}$$

Momentum

$$\rho \frac{\partial v}{\partial t} + \vec{v} \cdot \nabla v = -\nabla P + \nabla \cdot \vec{\tau} + \rho g \tag{2}$$

Energy

$$\frac{\partial (\rho H)}{\partial t} = -\nabla \cdot \left( \rho \vec{v} H \right) + \nabla \cdot (k \nabla T) + \frac{\partial P}{\partial t} \tag{3}$$

where $\rho$ is the density (kg/m$^3$), v is the velocity (m/s), $\vec{\tau}$ is the shear stress (Pa) $[\vec{\tau} = \mu \nabla \cdot \vec{v}]$, $\nabla P$ is the pressure gradient (Pa), g is the acceleration due to gravity (m/s$^2$), H is the total enthalpy (J/kg), k is the material thermal conductivity (W/m·K), T is the temperature (K), P is the pressure (Pa), and t is the time (s).

### 2.2. CFD Modelling

CFD is a tool that has gained momentum in studying and evaluating heat transfer in different systems and processes. One of the reasons is its ability to change operational conditions faster and more efficiently, which could be more complex and tedious in an experimental facility. As a result, the model could lead to changes in the geometry that would improve the system or process. The CFD simulations are developed in the commercial software called STAR-CCM+ v12.02 (Siemens, Germany).

#### 2.2.1. Geometrical Domain

The geometrical domain is crucial thanks to the importance of being the computational domain in which the simulation will be developed. The experimental facility to be represented is shown in Figure 1. The geometry is designed using Autodesk Inventor® v2017 and transformed into a CAD model.

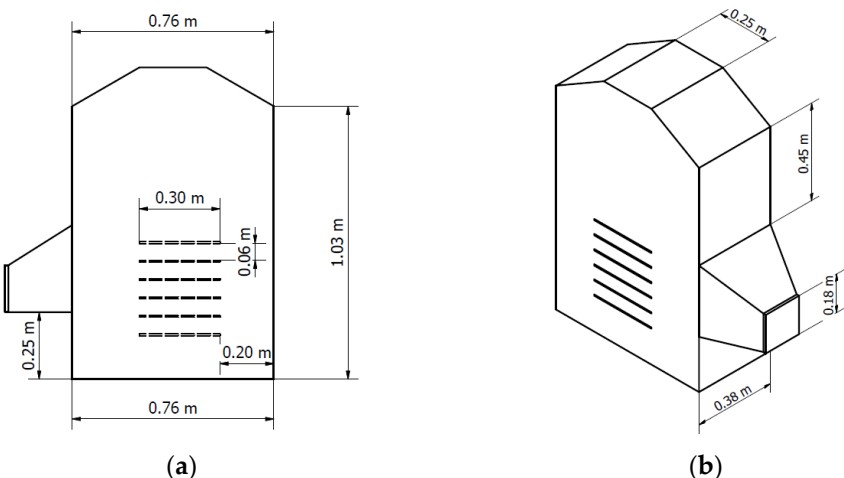

|           |           |
| :-------: | :-------: |
| (**a**)   | (**b**)   |

**Figure 1.** Dimensions of the convective drying chamber. (**a**) Lateral view and (**b**) Diagonal view.

### 2.2.2. Grid Generation

Grid generation is an essential step for the validation of the research project. Several mesh arrangements depend on the modelled system; the polyhedral-type mesh is used in this case. This type gives each cell many immediate neighbouring cells from which the software can obtain information and linear shape functions, resulting in a better approximation of the gradients, lower skewness angles, and a more accurate flux calculation than a tetrahedral mesh [40]. Another benefit of using a polyhedral-type mesh over other types for this particular application is that polyhedral cells allow for easy and gradual control of grid size changes (for coarsening or refining certain regions of the system). This avoids the sudden size changes resulting from using some trimmed hexahedral cells [40].

Figure 2 shows how the polyhedral mesh is implemented in the studied chamber. Figure 3 shows another approximation of the same chamber with a fruit sample represented by a disk. In this case, a refinement in the grid was made in the entrance, the outlet, and the disk, to have a more accurate resolution of the equations being solved.

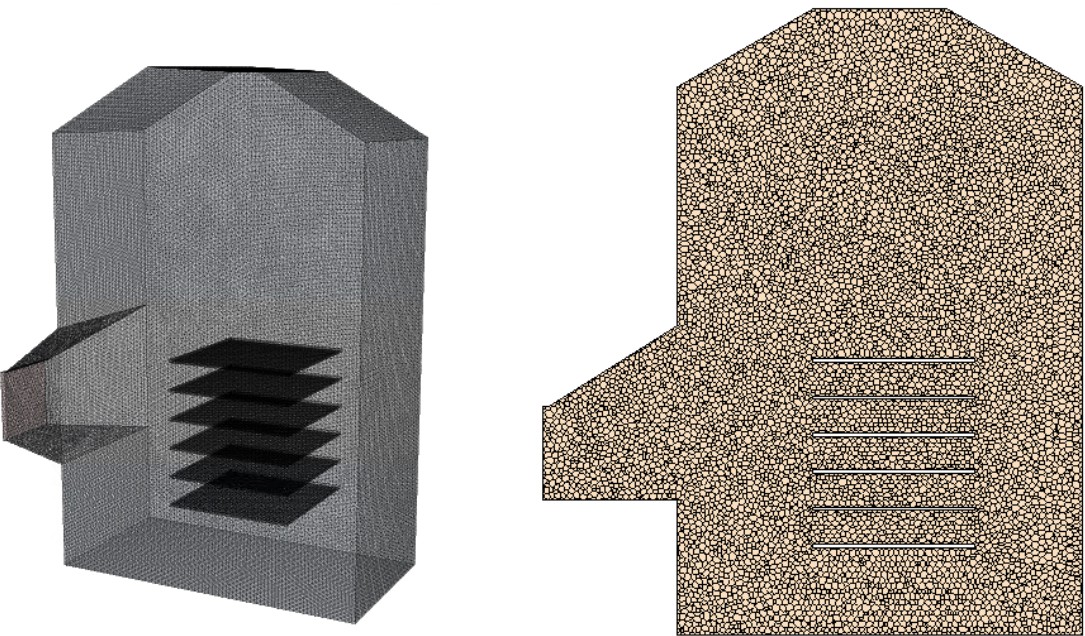

**Figure 2.** Visualisation of the polyhedral-type mesh in the studied chamber.

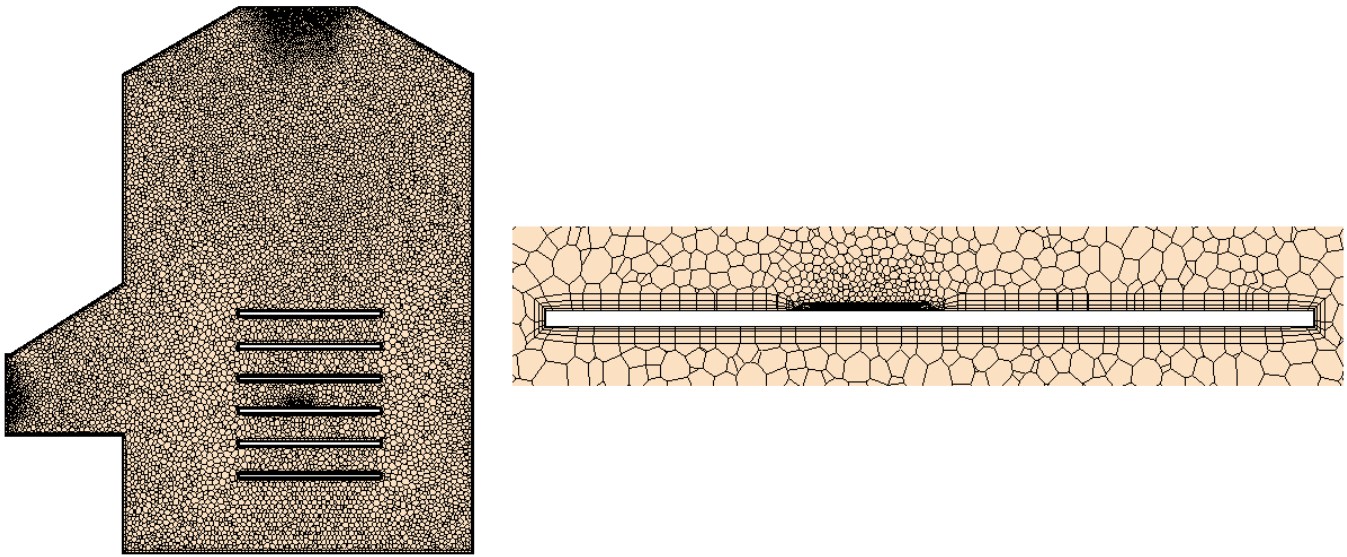

**Figure 3.** Visualisation of the grid in the studied chamber with a sample.

One key characteristic that must be considered is that the number of cells affects the computational time and the accuracy of the results. To balance these two variables, a mesh independence test was performed. The results of this test lead to a mesh selection.

### 2.2.3. Boundary and Initial Condition Selection

The specification of the Boundary and initial conditions of the system is a significant step in the pre-processing stage in CFD. For the first model, the inlet and outlet of the system are modelled as "velocity inlet" and "flow split outlet", respectively. These were chosen based on the behaviour of the system and the complex interactions that occur in it. The remaining spatial discretisation is modelled as a wall condition.

For the second implemented model, the inlet and outlet are modelled as "mass flow inlet" and "flow split outlet". In this model, the mass flow was used to have a more stable solution and correctly model the mass transfer in the system. As before, the remaining discretisation is modelled as a wall condition except for the disk, which is modelled as a solid. This condition is made to implement a mass transfer model.

Concerning the initial condition, this is defined for t $= 0$ s, when the temperature value is $T_0$ (300.15 K). The surface's boundary conditions are the equilibrium between convection and conduction heat.

$$h\left(T_s - T_{gb}\right) = (-k\nabla T) \tag{4}$$

where h is the convective heat transfer coefficient (W/m²·K), $T_{gb}$ is the air bulk temperature (K), $T_s$ is the surface temperature (K), and k is the material thermal conductivity (W/m·K).

The convective heat transfer coefficient is defined with the Chilton–Colburn relationship for turbulent flows [41] with Reynolds (Re), Prandtl (Pr), and Nusselt (Nu) numbers.

$$Nu = \frac{hL}{k} = 0.0296 Re^{\frac{4}{5}} Pr^{\frac{1}{3}} \tag{5}$$

where Re $= vL\rho/\mu$; Pr $= (C_p\,\mu)/k$; with $\rho$ is the density (kg/m³), v is the velocity (m/s), $\mu$ is the air viscosity (Pa·s), L is the air length path (m), $C_p$ is the specific heat (J/kg·K), k is the thermal conductivity (W/m·K), and h is the convective heat transfer coefficient (W/m²·K).

### 2.2.4. Physical Model Selection

An important step is the physical model selection. In this case, it is modelled as a transient system with a physical turbulence phenomenon exemplified by the k-ε model. This model decomposes the Navier–Stokes (NS) equations' instantaneous variables into their mean and fluctuations [42]. Complementary to this, the following models were also used: segregated model (which solves the momentum equations, one for each dimension [40]) and gravity. The air temperature and velocity are 313.15 K and 1 m/s, respectively.

The model k-ε was used for the turbulence model, represented by Equations (6) and (7). This turbulence model is mainly used in industrial applications [40]. It has also been proven that for this specific application, it does not have a significant difference from the k-ω turbulence model [43].

$$\frac{\partial(\rho k)}{\partial t} + \nabla \cdot \left(\rho k \vec{V}\right) = \nabla \cdot \left[\left(\mu + \frac{\mu_t}{\sigma_k}\right)\nabla k\right] + G_k - \rho\varepsilon \tag{6}$$

$$\frac{\partial(\rho\varepsilon)}{\partial t} + \nabla \cdot \left(\rho\varepsilon \vec{V}\right) = \nabla \cdot \left[\left(\mu + \frac{\mu_t}{\sigma_k}\right)\nabla\varepsilon\right] + C_{1\varepsilon}\frac{\varepsilon}{k}(G_k) - C_{2\varepsilon}\rho\frac{\varepsilon^2}{k} \tag{7}$$

where $G_k$ is the kinetic energy generation due to the velocity gradients' mean. The quantities $\sigma_k$ and $\sigma_\varepsilon$ are the Prandtl numbers for k and $\varepsilon$, respectively, with $C_{1\varepsilon}$ and $C_{2\varepsilon}$ as constants. $\mu_t$ is the turbulent viscosity (eddy viscosity) defined as:

$$\mu_t = \rho C_\mu \frac{k^2}{\varepsilon}. \tag{8}$$

with $C_{1\varepsilon} = 1.44$; $C_{2\varepsilon} = 1.92$; $C_\mu = 0.09$; $\sigma_k = 1.0$, and $\sigma_\varepsilon = 1.3$ [44–46] for turbulent airflow conditions in drying.

As specified before, the simulations must be transient to validate the CFD model. The necessary parameters specified in the model are time step, inner iterations per time step, and maximum physical time. The most critical parameter is the time step because diverse problems can appear if it is not correctly calculated. One of these problems is convergence when the time step is larger than the velocity magnitude. This causes intermediate points to be not solved, so that the following points have no previous solution while the CFD solver is expecting those solutions. This leads then to divergence. The CFL (Courant–Friedrichs–Lewy) condition is used to avoid the divergence problem. The recommended CFL values are below 0.1 to capture the interface accurately. The main results obtained from the simulations are temperature and velocity profiles.

After the first model is implemented, a second one is tested to study the mass transfer on the sample and the chamber. A multiphase model is used; the volume of Fluid (VOF) is the one selected in addition to a Fluid Film. The interaction of these two models will allow a chamber analysis from a mass transfer perspective.

The VOF model uses only one set of equations for the continuous phase, and the dispersed phase has a different equation for its volume fraction [45]. The continuity equation, which guarantees mass conservation, is described in Equation (9).

$$\frac{\partial\rho}{\partial t} + \frac{\partial\rho v_i}{\partial x_i} = 0 \tag{9}$$

where $v_i$ is the fluid velocity, t is the time, and $x_i$ is the spatial coordinate. In Equation (10), the momentum equation is described. This represents the Navier–Stokes equation.

$$\frac{\partial}{\partial t}\left(\rho v_j\right) + \frac{\partial}{\partial x_i}\left(\rho v_i v_j\right) = -\frac{\partial P}{\partial x_j} + \frac{\partial}{\partial x_i}\mu\left(\frac{\partial v_i}{\partial x_j} + \frac{\partial v_j}{\partial x_i}\right) + \rho g_j + F_j \tag{10}$$

In this equation, F, P, and g indicate external force per unit volume, pressure, and gravitational acceleration, respectively, $x_j$ is the spatial coordinate, $\mu$ is the dynamic viscosity,

and $v_j$ is the fluid velocity. The fluid properties are calculated as a function of the physical properties of each phase and their void fractions.

$$\rho_m = \sum_{i=1}^{n} \alpha_p * \rho_p \tag{11}$$

$$\mu_m = \sum_{i=1}^{n} \alpha_p * \mu_p \tag{12}$$

where $\rho_p$ is the density of the phase p and $\mu_p$ is the dynamic viscosity of the phase p. The variable $\alpha_p$ is the void fraction and can be expressed as Equation (13).

$$\frac{\partial \alpha_p}{\partial t} + u \nabla(\alpha_i) = 0. \tag{13}$$

## 3. Results and Discussion

### 3.1. Mesh Independence Test

Three meshes were tested to select the one that resulted in the best balance of the previously mentioned variables: computational time and solution accuracy. The number of cells in each mesh is 373,633, 792,269, and 2,178,418. Figure 4 presents a comparison between the three meshes with the variable temperature.

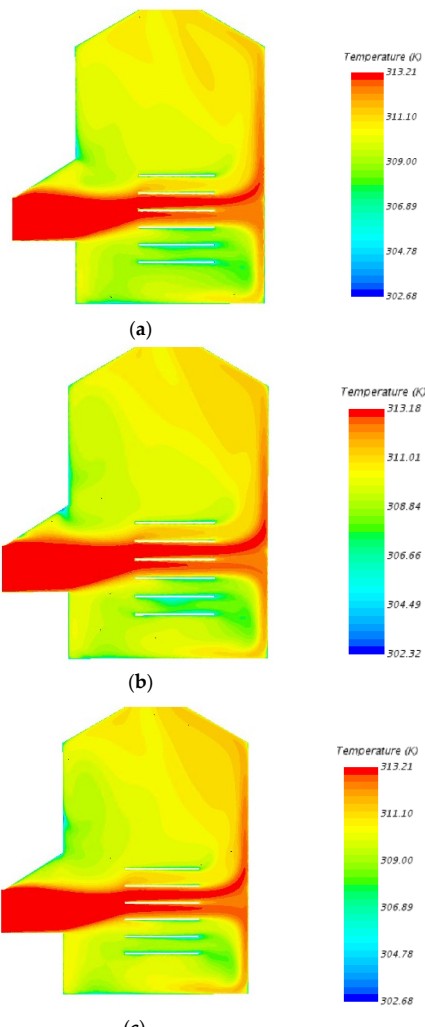

**Figure 4.** Comparison of the proposed meshes with a time of 45 s: (**a**) first mesh, (**b**) second mesh, and (**c**) third mesh.

As can be seen, the temperature results do not vary significantly. The volume average temperature was 308.37 K, 310.288 K, and 310.856 K calculated for meshes 1, 2, and 3, respectively. Nevertheless, an increase in computational time is seen. For the 45 s that the simulation was run on a 10-core computer, an estimated 51.4, 127.23, and 238.23 h were calculated for meshes 1, 2, and 3, respectively. With these results, the chosen mesh was the second one. The third one had a significant increase in computational time, while the first one had an unappropriated cell size for the total length of the chamber.

### 3.2. Model Validation

The data validation was made by comparing the experimental data at the exit of the chamber in a drying procedure and the simulation data in the exact location.

It is essential to mention that the data compared were only the ones corresponding to the time after the air temperature has stabilised since it includes food samples, and this affects the air temperature significantly at the beginning of the process due to changes in the relative humidity of the air, delaying the time to achieve temperature stabilisation. Comparison between experimental and simulated data are plotted in Figure 5. Due to the lack of samples and stability in relative humidity, the time to achieve a constant air temperature is noticeably faster than for the experimental data, as was explained before. It is also shown that the numerical data tend to have the same values as the experimental ones.

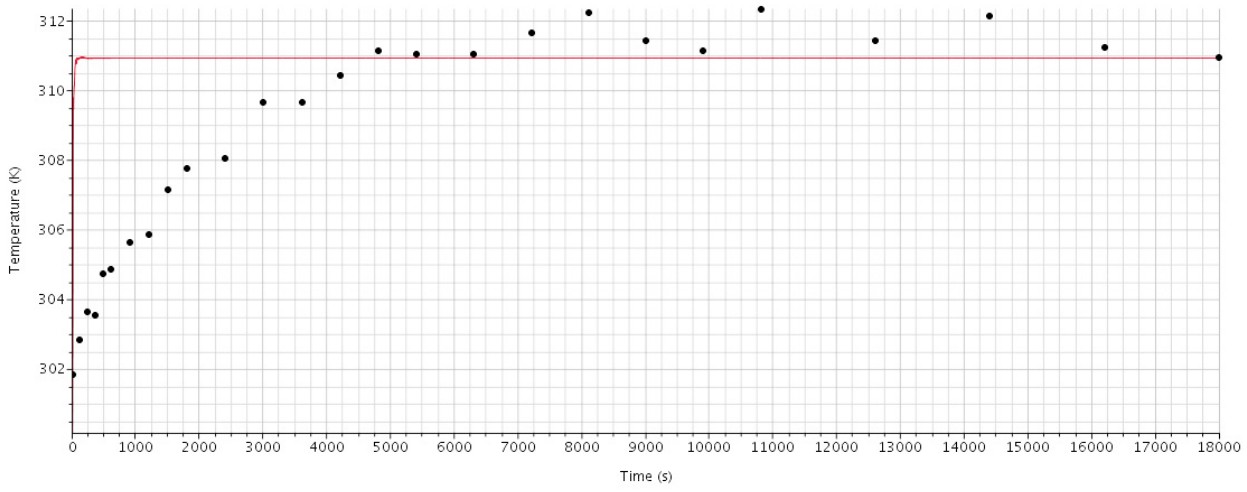

**Figure 5.** Comparison between experimental and numerical data.

### 3.3. Velocity Profile and Temperature Distribution

The results of the velocity profile can be seen in Figure 6. It can be observed that the air distribution does not present in a uniform manner. The velocity is optimum for the heat transfer by convection only in trays 2, 3, and 4 from top to bottom; also, stagnation occurs at the bottom and top of the chamber. This leads to reconsidering the tray positioning and the implementation of air distributors such as dampers and diffusers. In addition, it can be observed that the chamber has two recirculation areas, one at the bottom (below tray 4) and one at the top, near the air exit. As mentioned above, this can be advantageous if the air distribution is improved.

The temperature distribution is observed at different times in the solution in Figure 7. It shows that the temperature profile is not as uniform as expected. As well as in the velocity profile, only trays 2, 3, and 4 (from top to bottom) are receiving the required amount of heat needed to make the convective drying chamber successful. Another observation is the fact that the temperature starts to stabilise. The chamber will have the highest temperature in all domains in the first seconds. In contrast, the effective area (where the temperature reaches 40 °C, which is the optimal temperature) decreases slightly. The temperature starts to be

uniform at the top of the chamber and at the bottom, where it is not helpful, especially on trays 5 and 6, where drying will be less efficient, leading to uneven or poor product quality.

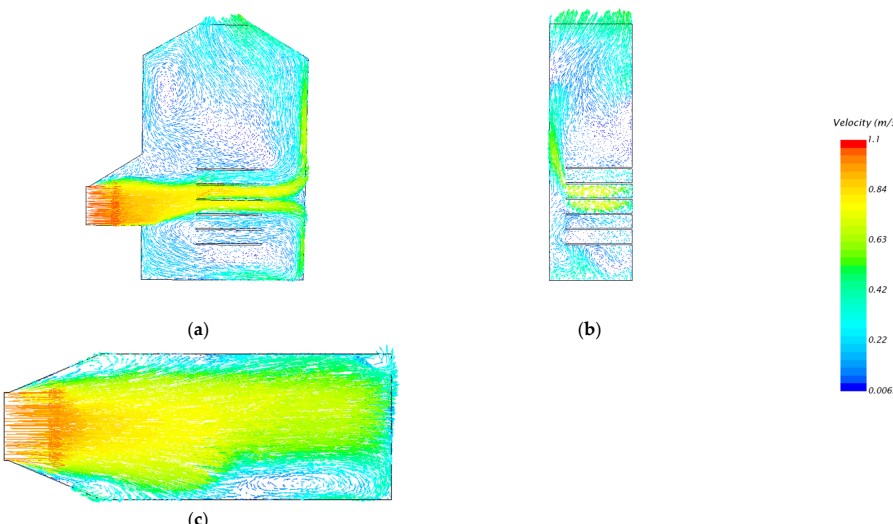

**Figure 6.** The velocity profile in the drying chamber is (**a**) normal to $X$, (**b**) normal to $Y$, and (**c**) normal to $Z$.

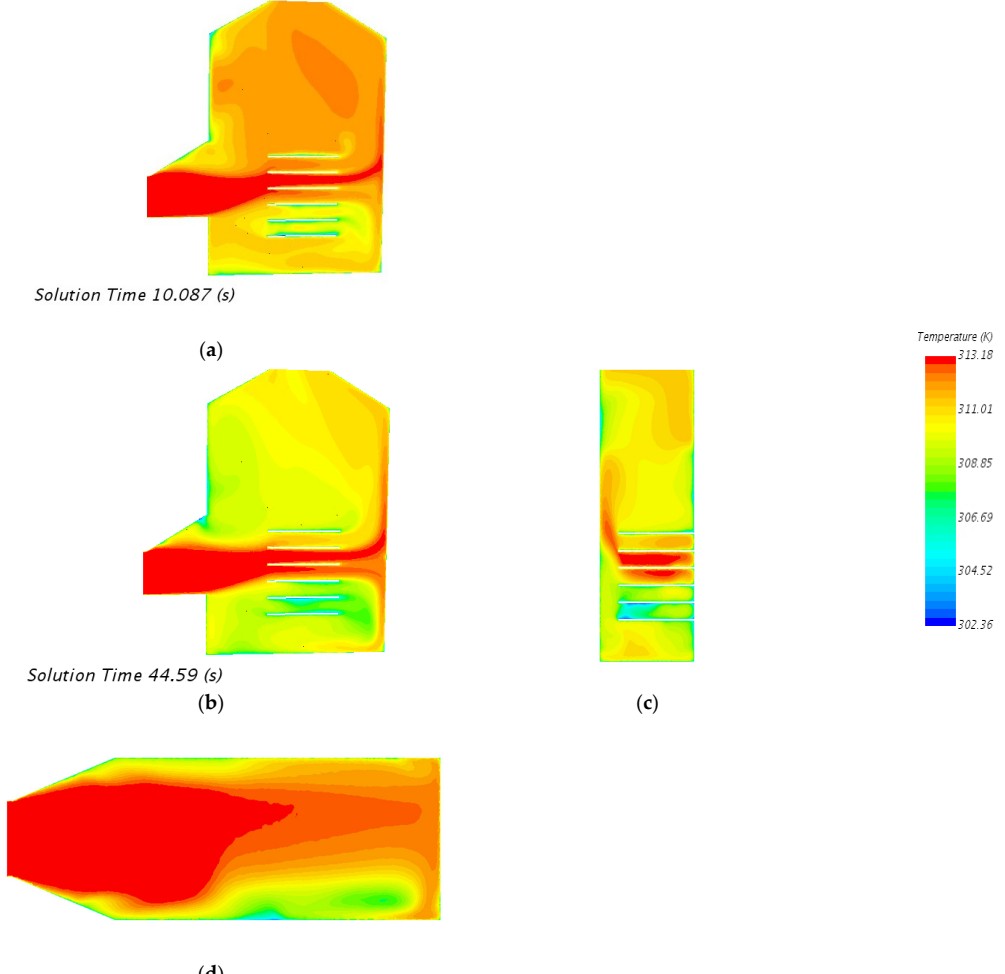

**Figure 7.** Temperature distribution at different times. (**a**) 10.087 s; (**b**) 44.59 s. At 44.59 s, it is also shown (**b**) normal to $X$, (**c**) normal to $Y$, and (**d**) normal to $Z$.

Air temperature and velocity are plotted in Figure 8. According to this, there is a relationship between these two factors that supports the behaviour observed in Figures 5 and 6; if stagnation of air increases (decreasing velocity), temperature drops, resulting in poor drying as is observed in the top and lower regions in Figures 6 and 7. This implies uneven quality products or, even worse, increasing production time to reprocess or dry the product.

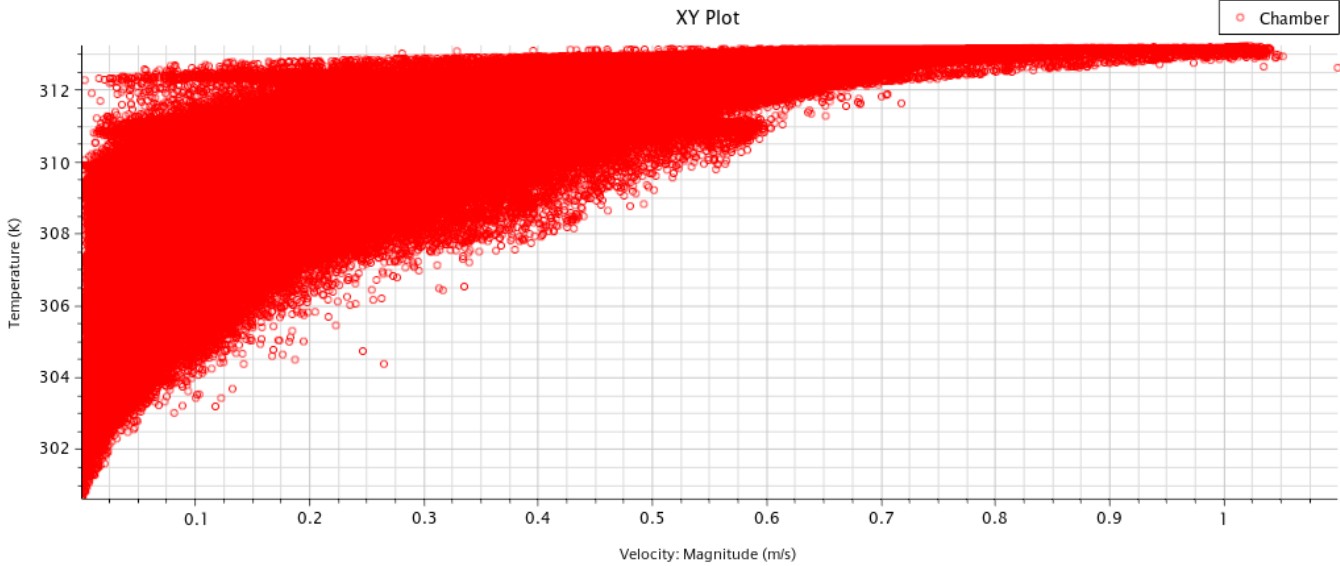

**Figure 8.** Velocity vs. temperature in the drying chamber.

Air distribution is a characteristic that must be assessed to improve drying and achieve a uniform distribution. This has been done by researchers, such as Mathioulakis et al. [47], who assessed the relationship between air velocity and weight loss on drying and agreed with observations made in this study as well as the work by Amanlou and Zomorodia [48]. They proposed different designs to improve the drying process. Other studies have analysed the distribution and airflow [47,49], defining air velocity thresholds with the effect on the drying rate, reassuring the crucial role of air velocity in air drying efficiency, as is observed in this study from a different point of view in relation with the air velocity and temperature distribution.

One of the observations made in their work is that low air velocity and poor air distribution imply a poor design and, ergo, wrong or more prolonged operation, which is seen in the results of this study.

### 3.4. Velocity and Mass Profile and Temperature Distribution

Nevertheless, the results of the first part gave an insight into how the chamber must be modified; the second model was implemented. In this case, three results are achieved—mass transfer, velocity profile, and temperature distribution within the chamber. As was expected, only certain parts of the chamber achieve the desired temperature (40 °C), and as a result, only a part has the desired drying effect.

Figure 9 shows the temperature achieved by the chamber sample in the chamber. This temperature is adequate to have a good drying operation. This was achieved by the position in which the sample was located. However, the same results can be seen for this new model. The distribution of the air must be improved.

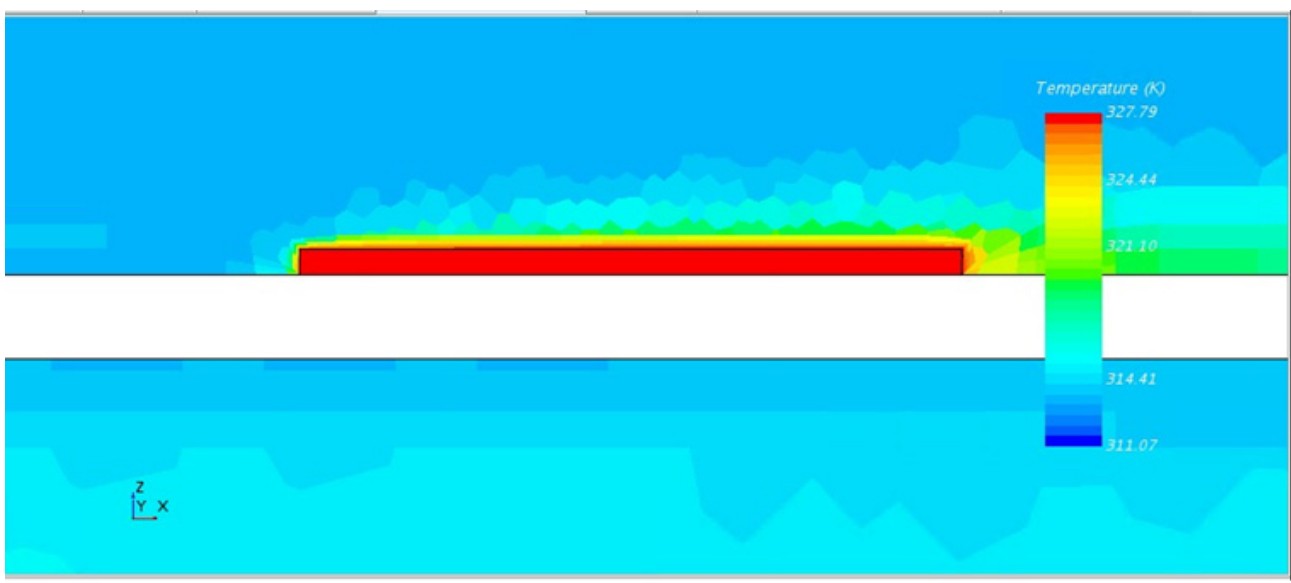

**Figure 9.** The temperature of the sample.

In Figure 10, it can be seen how the fraction of water vapour decreases with time. This proves that the chamber is working, but changes are needed, as shown in Figure 11.

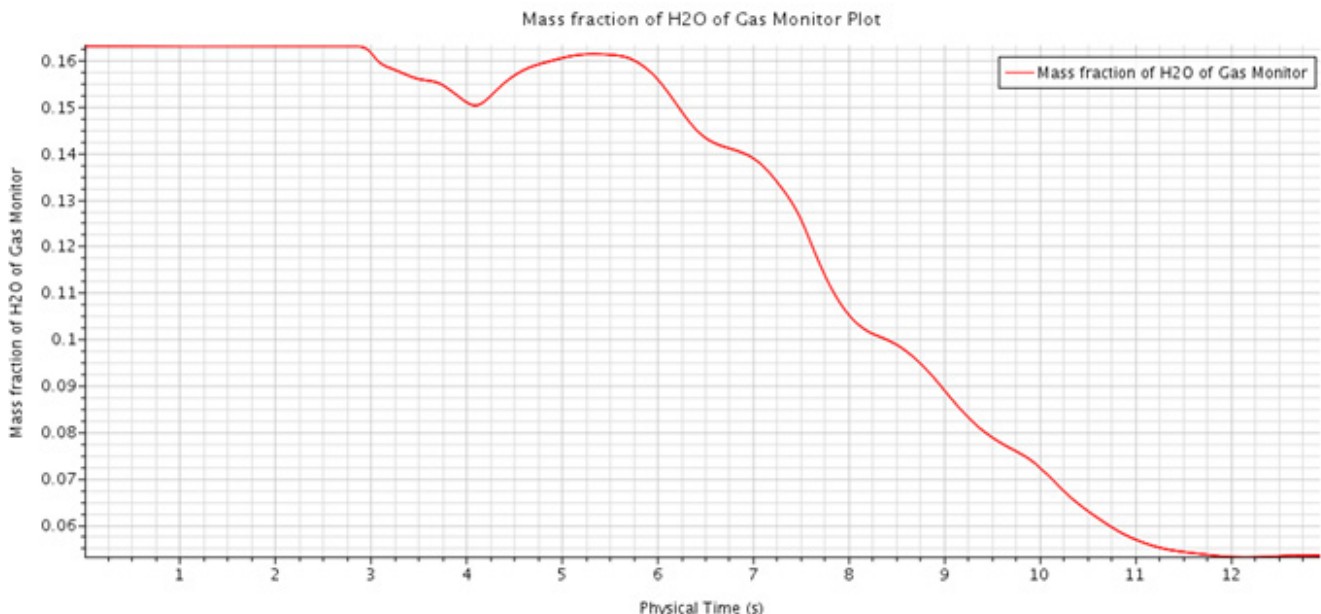

**Figure 10.** Mass fraction of water vapour.

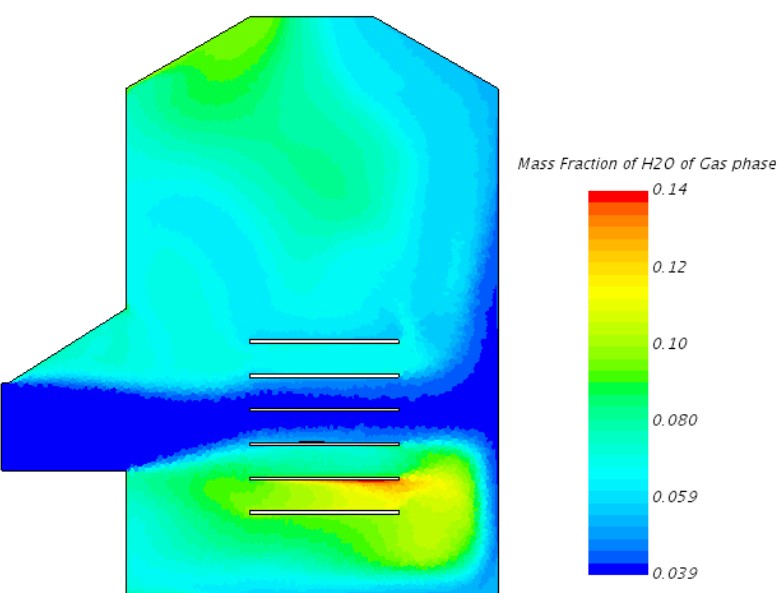

**Figure 11.** Mass fraction of water vapour distribution.

## 4. Conclusions

A CFD simulation was implemented using a transient system model with the k-ε model for turbulence and the CFL criterion to understand the convective drying process. This is a handy and powerful tool to achieve the desired assessment to identify the temperature and air profiles in the convective drying chamber. The results of the air distribution show that there are two specific areas of low air velocity that tend to create recirculation. They are located at the air exit and the other at the bottom of the chamber, which results in poor drying.

Besides this, the temperature profile shows only a homogeneous drying in half of the trays (2, 3, and 4), reducing drying efficiency drastically, especially for the bottom ones. Air distributors or dampers are necessary to deliver and improve the convective chamber. The study has concluded that a redesign is needed to achieve the expected results from the system. This assessment is being executed in another project, and the simulations are still running.

Simulation using CFD has been demonstrated to be a reliable optimisation tool to avoid unnecessary and expensive experiments to improve the design and can be used in subsequent research to predict the drying time—with the additional help of a thin film model implementation or a more extensive heat and mass transfer model.

**Author Contributions:** Conceptualization, M.A.M.D.-G., N.R., C.A.G.V. and J.C.G.D.; methodology, M.A.M.D.-G. and C.A.G.V.; software, M.A.M.D.-G. and C.A.G.V.; validation M.A.M.D.-G. and C.A.G.V.; formal analysis, M.A.M.D.-G., N.R., C.A.G.V. and J.C.G.D.; investigation, M.A.M.D.-G., N.R., C.A.G.V. and J.C.G.D.; resources N.R. and J.C.G.D.; data curation, M.A.M.D.-G. and C.A.G.V.; writing—original draft preparation, M.A.M.D.-G. and C.A.G.V.; writing—review and editing, N.R. and J.C.G.D.; visualisation, M.A.M.D.-G. and C.A.G.V.; supervision, N.R. and J.C.G.D.; project administration N.R. and J.C.G.D.; funding acquisition, N.R. and J.C.G.D. All authors have read and agreed to the published version of the manuscript.

**Funding:** This research received no external funding.

**Data Availability Statement:** Not applicable.

**Acknowledgments:** We want to thank the conjoint effort between Universidad de Los Andes and Universidad del Valle for the opportunity to work on this project. It is a great chance to exchange knowledge and develop a model that can help develop efficient equipment for the drying industry.

**Conflicts of Interest:** The authors declare no conflict of interest.

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
