# Peer review of "3D Computational Fluid Dynamics Analysis of a Convective Drying Chamber"

_processes, doi:10.3390/pr10122721_

Round 1
Reviewer 1 Report
This paper gives a clear explanation about the benefits of using Computational fluid dynamics (CFD) in a drying chamber. The problem of low air velocity and poor air distribution was identified. The areas found for air recirculation in the chamber were at the bottom and the air exit. It finds a result of a low-cost substitute to identify air and temperature profiles.
Reviewer 2 Report
In this manuscript, the authors present the development of a mathematical model through Computational Fluids Dynamics (CFD) and are to implement air distributors or modify the tray positioning to make the drying air and temperature distribution homogeneous. There are good application and science value. However, there are some significant issues that should be carefully analysed.
1)The units and letters of density are wrong in Line 81.
2) The units of the acceleration due to gravity are wrong in Line 82.
3) The units of density are wrong in Line 140.
4) The expression of “?_?” is wrong in Line 140.
5) The font format of " Bactris guineensis" should be in italics in Line 338.
6) The font format of " Cornus mas L." should be in italics in Line 346. The linguistic term must be in italics. The author should check for similar errors and make corrections.
7) The initial letter format of article names in references is not uniform, such as Ref. [24], in Line 361.
Reviewer 3 Report
See attached document.

Round 2
Reviewer 3 Report
The authors mostly removed the mentioned remarks and improved the text.
However, there are still minor shortcomings as stated below:
Table 1 – no data source defined.
Subscripts in expressions are usually written vertically (not italics), if they are not variables.
The literature has been supplemented, but not enough. There is also more recent literature.
